



**Intercomparison of four methods to estimate coral calcification under various**
**environmental conditions**
Miguel Gómez Batista[1], Marc Metian[2], François Oberhänsli[2], Simon Pouil[2], Peter W.
Swarzenski[2], Eric Tambutté[3], Jean-Pierre Gattuso[4,5], Carlos M. Alonso Hernández[1], Frédéric
Gazeau[4]
[1]Centro de Estudios Ambientales de Cienfuegos, Cuba
[2]International Atomic Energy Agency, Environment Laboratories, 4a Quai Antoine 1er, MC-
98000 Monaco, Principality of Monaco
[3]Centre Scientifique de Monaco, Department of Marine Biology, MC-98000 Monaco,
Principality of Monaco
[4]Sorbonne Université, CNRS, Laboratoire d'Océanographie de Villefranche, LOV, F-06230
Villefranche-sur-Mer, France
[5]Institute for Sustainable Development and International Relations, Sciences Po, 27 rue Saint
Guillaume, F-75007 Paris, France
Correspondence to: Miguel Gómez Batista (mgomezbatista@gmail.com)
Keywords: Calcification; Coral; Alkalinity anomaly; Calcium anomaly; [45]Ca incorporation;
[13]C incorporation





## 22    Abstract

Coral reefs are constructed by calcifiers that precipitate calcium carbonate to build
their shells or skeletons through the process of calcification. Accurately assessing coral
calcification rates is crucial to determine the health of these ecosystems and their response to
major environmental changes such as ocean warming and acidification. Several approaches
have been used to assess rates of coral calcification but there is a real need to compare these
approaches in order to ascertain that high quality and intercomparable results can be
produced. Here, we assessed four methods (total alkalinity anomaly, calcium anomaly, $^{45}$Ca
incorporation and $^{13}$C incorporation) to determine coral calcification of the reef-building coral
*Stylophora pistillata*. Given the importance of environmental conditions on this process, the
study was performed under two pH (ambient and low level) and two light (light and dark)
conditions. Under all conditions, calcification rates estimated using the alkalinity and calcium
anomaly techniques as well as $^{45}$Ca incorporation were highly correlated. Such a strong
correlation between the alkalinity anomaly and $^{45}$Ca incorporation techniques has not been
observed in previous studies and most probably results from improvements described in the
present paper. The only method which provided calcification rates significantly different from
the other three techniques was $^{13}$C incorporation. Calcification rates based on this method
were consistently higher than those measured using the other techniques. Although reasons
for these discrepancies remain unclear, the use of this technique for assessing calcification
rates in corals is not recommended without further investigations.



## 1. Introduction

Calcification is the fundamental biological process by which organisms precipitate

calcium carbonate. Calcifying organisms take up calcium and carbonate or bicarbonate ions to
build their biomineral structures (aragonite, calcite and/or vaterite) which have physiological,
ecological and biogeochemical functions. Moreover, calcium carbonate plays a major role in
the services provided by ecosystems to human societies.

The ocean has absorbed large amounts of anthropogenic $CO_2$ since the start of the

industrial revolution and is currently sequestering about 22% of $CO_2$ emissions (average
2008-2017; Le Quéré et al., 2018). This massive input of $CO_2$ in the ocean impacts seawater
chemistry with a decrease in seawater pH, carbonate ion concentrations $[CO_3^{2-}]$ and an
increase in $CO_2$ and bicarbonate concentrations $[HCO_3^-]$. These fundamental changes to the
carbonate system are referred to as "ocean acidification" (OA; Gattuso and Hansson, 2011).
Models project that the average surface water pH will drop by 0.06 to 0.32 pH units by the
end of the century (IPCC, 2014).

The effect of OA on the ocean is currently the subject of intense research with

particular attention to organisms producing $CaCO_3$. For instance, coral communities have
already proven to be particularly vulnerable to rapidly changing global environmental
conditions (e.g. Albright et al., 2018). In order to help project the future of coral reefs,
accurate estimates of calcification rates during realistic perturbation experiments are
necessary in order to produce high quality and intercomparable results (Langdon et al., 2010).

Several methods are available to quantify rates of coral calcification. Calcification can

be measured as the increase of $CaCO_3$ mass (e.g. the buoyant weight technique; Jokiel et al.,
1978) or following the incorporation of radio-labelled carbon or calcium in the skeleton



(Goreau, 1959), but also through the quantification of changes in a seawater constituent that is
stoichiometrically related to the amount of $CaCO_3$ precipitated. For instance, the alkalinity
anomaly technique (Smith and Key, 1975) has been widely used to estimate net calcification
of organisms and communities, especially of corals and coral reef environments (e.g. Smith
and Kinsey, 1978; Gazeau et al., 2015; Albright et al., 2016; Cyronak et al., 2018). Total
alkalinity ($A_T$) is directly influenced by bicarbonate and carbonate ion concentrations together
with a multitude of other minor compounds (Wolf-Gladrow et al., 2007). Calcification
consumes carbonate or bicarbonate, following the reversible reaction:
$Ca^{2+} + 2HCO_3^- \leftrightarrow CaCO_3 + CO_2 + H_2O$                                        (1)

Calcification consumes two moles of $HCO_3^-$, hence decreasing $A_T$ by two moles per

mole of $CaCO_3$ produced (eq. 1). It is possible to derive the rate of net calcification (gross
calcification - dissolution) by measuring $A_T$ before and after incubating an organism or a
community. This method must assume, however, that only calcification influences $A_T$ (Smith
and Key, 1975).

In contrast to $A_T$, the concentration of calcium ($Ca^{2+}$) in seawater is only biologically

influenced by net calcification and a 1:1 relationship can be used to derive net calcification
rates (eq. 1). The depletion of $A_T$ and $Ca^{2+}$ needs to be corrected for gains of $A_T$ and $Ca^{2+}$
resulting from evaporation. These corrections can be applied through the incubation of
seawater in the absence of coral (Schoepf et al., 2017). Both the alkalinity anomaly and
calcium anomaly methods are non-destructive and typically show a solid agreement
(Chisholm and Gattuso, 1991; Murillo et al., 2014; Gazeau et al., 2015).

The $^{45}Ca$ incorporation technique has been used since the 1950's (Goreau and Bowen,

1955; Goreau, 1959). While earlier techniques showed low reproducibility, methodological
improvements led to a significant reduction of the deviations between replicates (see





Tambutté et al., 1995, for more details). The strength of this method is that it is extremely
sensitive for measuring short-term variations in gross calcification rates. However, in contrast
to the $A_T$ and $Ca^{2+}$ anomaly techniques, it is a sample-destructive method.

Previous studies designed to compare calcification rate estimates using the $^{45}Ca$

incorporation and $A_T$ anomaly methods revealed subtle discrepancies. For example, Smith and
Roth in Smith and Kinsey (1978) reported an overestimation of rates based on the $^{45}Ca$
method. In contrast, Tambutté et al. (1995) and Cohen et al. (2017) reported a decrease in $A_T$
without concomitant incorporation of $^{45}Ca$, therefore suggesting an overestimation of
calcification derived from $A_T$ measurements. However, during these studies, in order to avoid
radioactive contamination of laboratory equipment, estimates of calcification were not
performed during the same incubations, but rather during incubations performed over two
consecutive days.

In contrast to the $^{45}Ca$ incorporation method, to the best of our knowledge, no studies

have used carbon-based incorporation techniques to estimate coral calcification rates in the
framework of ocean acidification. Past studies that compared carbon and calcium
incorporation rates in coral skeletons based on a double labelling technique with $H^{14}CO_3$ and
$^{45}Ca$ showed that only a minor proportion of the labelled seawater carbon is incorporated in
the skeleton (e.g. Marshall and Wright, 1998) and that the major source of dissolved inorganic
carbon for calcification is metabolic $CO_2$ (70–75% of the total $CaCO_3$ deposition; Furla et al.,
2000). Consequently, under both light and dark conditions, the rate of $^{45}Ca$ deposition appears
greater than the rate of $^{14}C$ incorporation (Furla et al., 2000). To the best of our knowledge,
only one study estimated calcification rates of a benthic calcifier (coralline algae) using a
stable carbon isotopic technique through addition of $^{13}C$-labelled bicarbonate (McCoy et al.,

2016).





113   The present study aimed at comparing calcification rates measured using the alkalinity

114 and calcium anomaly methods, as well as the $^{45}$Ca and $^{13}$C incorporation techniques.



## 2. Material and Methods


Colonies of the reef-building coral *Stylophora pistillata* were incubated in the
laboratory, both in the light and dark, under ambient and lowered pH conditions. At ambient
pH (experiment conducted in July-August 2017), two sets of incubations were performed
using either $^{45}Ca$ or $^{13}C$ additions and calcification rates based on these techniques were
compared to those derived, during the same incubations, by the alkalinity and calcium
anomaly techniques. At lowered pH (experiment conducted in August 2018), no incubations
with $^{13}C$ addition were conducted and only the three other techniques were compared.

### 2.1. Biological material and experimental set-up


Specimens used in this experiment originated from colonies of the coral *Stylophora*
*pistillata* (Esper 1797) initially sampled in the Gulf of Aqaba (Red Sea, Jordan) and
transferred to the Scientific Centre of Monaco where they were cultivated under controlled
conditions for several years. In June 2017, terminal branches of *S. pistillata*, free of boring
organisms, were cut and suspended with a nylon line to allow tissues to fully cover the
exposed skeleton for at least five weeks (Tambutté et al., 1995; Houlbrèque et al., 2015). The
nubbins were fed with rotifers (once a day) and artemia nauplii (twice a week) and kept under
an irradiance of 200 µmol photons m$^{-2}$ s$^{-1}$ (12:12 light:dark photoperiod, light banks: HQI
250W Nepturion - BLV (Germany) / 200 µmol photons m$^{-2}$ s$^{-1}$), a seawater temperature of 25
$\pm$ 0.5 °C and a salinity of 38 $\pm$ 0.5. Before the start of the experiment, specimens were
transferred to the International Atomic Energy Agency (IAEA). For the second set of
experiments in 2018, nubbins were prepared in June 2018 and cultured, under the conditions
described above, at IAEA except that colonies were fed twice a week with newly hatched



brine shrimp nauplii (1 nauplius ml$^{-1}$). Biometrics parameters (size, weight) on the biological
material are shown in Table 1.

Different types of incubations were conducted. In July-August 2017, one set of

incubations was performed under ambient pH conditions with the addition of radioactive
calcium dichloride ($^{45}$CaCl$_2$). During the same period, another set of incubations was
performed, under ambient pH conditions, with addition of labelled $^{13}$C-sodium bicarbonate
($^{13}$C-NaHCO$_3$ 99%). Finally, in August 2018, one set of incubations was performed under
lowered pH conditions (see thereafter for more details) with the addition of $^{45}$CaCl$_2$. For all
sets of incubations, organisms were incubated for 5 to 11 hours (Table 1), both in the light
and dark, in 500 mL polyethylene beakers equipped with a magnetic stirrer (Fig. 1). Six and
five replicates were used, respectively, at ambient a low pH. Furthermore, for all sets of
incubations, one beaker was incubated, under the same conditions as the other beakers,
without coral and served as a control.

For each set of incubations, 2.4 L of seawater, pumped continuous from offshore of

the IAEA Monaco premises at 50 m depth, were filtered onto 0.2 μm (GF/F, 47 mm). For
incubations performed at lowered pH condition, pure CO$_2$ was bubbled in the 2.4 L initial
seawater batch using an automated pH-stat system (IKS Aquastar©) until the target pH was
reached. The pH electrode from the pH-stat system was inter-calibrated using a glass
combination electrode (Metrohm, Ecotrode Plus) calibrated on the total scale using a TRIS
buffer solution with a salinity of 35 (provided by A. Dickson, Scripps Institution of
Oceanography, San Diego). Initial pH$_T$ (total scale) levels were set to ~7.2. It must be stressed
that pH levels were not regulated during the incubations. For $^{45}$Ca-incubations, this initial
batch was spiked with ca. 10 μL of $^{45}$CaCl$_2$ to reach a nominal activity of 25 Bq mL$^{-1}$. Before
distributing seawater to the experimental beakers, a one-milliliter aliquot of seawater was



removed for the precise determination of the initial activity. Samples were stored, in the dark,
in high-performance glass vials for 24 h before counting. For $^{13}$C-incubations, to determine
seawater background isotopic level ($\delta^{13}$C) of the dissolved inorganic carbon pool ($\delta^{13}$C-$C_T$),
three 27 mL samples were collected and gently transferred to glass vials avoiding bubbles.
Then, ~8.95 mg of $^{13}$C-NaHCO$_3$ were added to the batch of filtered ambient seawater to
increase $\delta^{13}$C-$C_T$ to ca. 1,500‰. For the determination of $\delta^{13}$C-$C_T$ after enrichment, two 27
mL samples were handled as described above. The vials were then sealed after being
poisoned with 10 µL of saturated mercuric chloride (HgCl$_2$) and stored upside-down at room
temperature in the dark for subsequent analysis.

For all sets of incubations, samples for the measurements of pH$_T$, $A_T$ (200 mL), and

Ca$^{2+}$ concentrations (50 mL) were taken before distributing seawater to the experimental
beakers. While pH$_T$ was measured immediately after sampling, samples for $A_T$ measurements
were poisoned with 40 µL of 50% saturated HgCl$_2$ and stored in the dark at 4 °C pending
analysis less than two weeks later. Samples for [Ca$^{2+}$] measurements were not poisoned and
stored in the dark at 4 °C pending analysis less than two weeks after sampling.

Gravimetrically determined amounts of filtered seawater (ca. 300 g) were transferred

to the incubation containers which were placed in a temperature-controlled (IKS Aquastar©)
water bath maintained at 25 ± 0.5 °C. Coral nubbins were suspended with a nylon line in the
experimental beakers 4 cm below the water level covered with transparent film to limit
evaporation (Fig. 1). During the low pH incubations conducted in 2018, to avoid a
physiological stress, coral nubbins were acclimated by gradually lowering pH to the target
levels during 24 h. This acclimation was performed in an open-flow 20 L aquarium (one full
water renewal per hour) using a pH-stat system as previously described and with a pH
decrease of ca. 0.03 units h$^{-1}$.



Incubations in the light were performed at an irradiance of 200 µmol photons $m^{-2}$ $s^{-1}$

during daytime whereas dark incubations were conducted at night. Before the beginning of

the incubations, all beakers (containing corals) were precisely weighed at ± 0.01 g (Sartorius

BP 310S).

At the conclusion of the incubations, all beakers were precisely weighed to evaluate

evaporation and seawater samples were analyzed for $pH_T$, $A_T$ and $[Ca^{2+}]$ as well as for $^{45}Ca$

activity or $\delta^{13}C$-$C_T$ depending on the type of incubations. $pH_T$ was measured immediately and

samples for $A_T$ and $[Ca^{2+}]$ determinations were filtered onto 0.2 µm (GF/F, Ø 47 mm),

poisoned with saturated $HgCl_2$ (only for $A_T$) and stored in the dark at 4 °C pending analysis

(within two weeks). The corals were then removed from the beakers for the analysis of

incorporated $^{45}Ca$ or $^{13}C$. Three additional corals which were not incubated were processed

for carbon isotopic composition of the previously accreted calcium carbonate (see section

"2.3. Computations and statistics").

## 2.2. Analytical techniques

Immediately after sampling, $pH_T$ was measured on a Metrohm 826 mobile pH-logger

and a glass electrode (Metrohm, Ecotrode Plus) calibrated on the total scale using a TRIS
buffer of salinity 35 (provided by A. Dickson, Scripps University, USA). $A_T$ was determined
in triplicate 50 mL subsamples by potentiometric titration on a titrator Titrando 888
(Metrohm) coupled to a glass electrode (Metrohm, Ecotrode Plus) and a thermometer
(pt1000). The pH electrode was calibrated before every set of measurements on the total scale
using a TRIS buffer of salinity 35 (provided by A. Dickson, Scripps University, USA).
Measurements were carried out at a constant temperature of 25 °C and $A_T$ was calculated as
described in Dickson et al. (2007). Certified reference material (CRM; batches 143 and 156)





provided by A. Dickson (Scripps University, USA) were used to check precision (standard
deviation within measurements of the same batch) and accuracy (deviation from the certified
nominal value). Over the six series of $A_T$ measurements performed during the experiment,
mean accuracy and precision (± SD) were respectively 7.2 ± 1.2 and 1.2 ± 0.2 µmol kg$^{-1}$.
[$Ca^{2+}$] was determined in triplicate using the ethylene glycol tetra acetic acid (EGTA)
potentiometric titration (Lebel and Poisson, 1976). About 10 g of sampled seawater and 10 g
of $HgCl_2$ solution (ca. 1 mmol L$^{-1}$) were accurately weighed out. Then, about 10 g of a
concentrated EGTA solution (ca. 10 mmol L$^{-1}$, also by weighing) was added to completely
complex $Hg^{2+}$ and to complex nearly 95% of $Ca^{2+}$. After adding 10 mL of borate buffer
(pH$_{NBS}$ ~ 10) to increase the pH of the solution, the remaining $Ca^{2+}$ was titrated by a diluted
solution of EGTA (ca. 2 mmol L$^{-1}$) using a tritrator (Titrando 888, Metrohm) coupled to an
amalgamated silver combined electrode (Metrohm Ag Titrode). Following Cao and Dai
(2011), the volume of EGTA necessary to titrate the remaining ca. 5% of $Ca^{2+}$ were obtained
by manually fitting a polynomial function to the first derivative of the titration curve using the
function "loess" of the R software[1]. The EGTA solution was calibrated prior to each
measurement series using International Association for the Physical Sciences of the Oceans
(IAPSO) standard seawater (salinity = 38.005). Mean [$Ca^{2+}$] precision obtained using this
technique was 2.9 µmol kg$^{-1}$ (n = 40), corresponding to a coefficient of variation (CV) of

0.026%.

To determine the specific activity in radio-labelled seawater, the 1 mL aliquots were

transferred to 20 mL glass scintillation vials and mixed in proportion 1:10 (v:v) with
scintillation liquid Ultima Gold ™ XR. According to a method adapted from Tambutté et al.
(1995), at the end of incubation sampled nubbins were immersed for 30 min in beakers

---

[1]The R Development Core Team, R.: A language and environment for statistical computing, 2018.

containing 300 mL of unlabelled seawater to achieve isotopic dilution of the $^{45}$Ca contained in
the gastrovascular cavity. Constant water motion was provided in the efflux medium by
magnetic stirring bars. Tissues were then dissolved completely in 1 mol L$^{-1}$ NaOH at 90 °C
for 20 min. The skeleton was rinsed twice in 1 mL NaOH and twice in 5 mL in MilliQ water.
It was then dried for 72 h at 60 °C, weighed (referred thereafter to as skeleton dry weight),
and dissolved in 12 N HCl. Three 200 μL aliquots from each skeleton dissolution were
transferred to 20 mL glass scintillation vials and mixed with 10 mL scintillation liquid Ultima
Gold $^{TM}$ XR. Radioactive samples were thoroughly mixed to homogenize the solution and
kept in the dark for 24 h before counting. The radioactivity of $^{45}$Ca was counted using a Tri-
Carb 2900 Liquid Scintillation Counter. Counting time was adapted to obtain a propagated
counting error of less than 5% (maximal counting duration was 90 min). Radioactivity was
determined by comparison with standards of known activities and measurements were
corrected for counting efficiency and physical radioactive decay.

The analyses of seawater $\delta^{13}$C-$C_T$ as well as of the $^{13}$C signature of coral calcified

tissues were performed at Leuven University. For $\delta^{13}$C-$C_T$ analyses, a helium headspace (5
mL) was created in the vials and samples were acidified with 2 mL of phosphoric acid
(H$_3$PO$_4$, 99%). Samples were left to equilibrate overnight to transfer all $C_T$ to gaseous $CO_2$.
Samples were injected in the carrier gas stream of an EA-IRMS (Thermo EA1110 and Delta
V Advantage), and data were calibrated with NBS-19 and LSVEC standards (Gillikin and
Bouillon, 2007). Corals were treated following the same protocol as for $^{45}$Ca incorporation
measurements and powdered. Triplicate subsamples of carbonate powder (~100  μg) were
placed into gas-tight vials, flushed with helium, and converted into $CO_2$ with H$_3$PO$_4$. After 24
h, subsamples of the released $CO_2$ were injected into the EA-IRMS system as described
above. Data were calibrated with NBS-19 and LSVEC. Carbon isotope data are expressed in





the delta notation (δ) relative to Vienna Pee Dee Belemnite (VPDB) standard and were
calculated as:
$R_{sample} = \frac{\delta^{13}C_{sample}}{1000 + 1} \cdot R_{VPDB}$                                          (2)

## 258    2.3. Computations and statistics

The carbonate chemistry was assessed using $pH_T$ and $A_T$ and the R package seacarb[2].

Propagation of errors on computed parameters was performed using the new function "error"
of the package seacarb (Orr et al., 2018) on the R software, considering errors associated to
the estimation of $A_T$ as well as errors on dissociation constants.

Estimates of coral calcification rates based on changes in $A_T$ and $[Ca^{2+}]$ during

incubations were computed following equations (3) and (4), respectively. As shown in these
equations, initial levels of $A_T$ and $[Ca^{2+}]$ are not necessary to compute calcification rates and
only final values in the incubations with corals and without corals (controls) were used:
$G_{AT} = -\frac{(A_{T2} - A_{T1}) - (A_{T2c} - A_{T1})}{2t} \cdot \frac{W_w}{W_c} = -\frac{(A_{T2} - A_{T2c})}{2t} \cdot \frac{W_w}{W_c}$          (3)
$G_{Ca} = -\frac{(Ca_2 - Ca_1) - (Ca_{2c} - Ca_1)}{t} \cdot \frac{W_w}{W_c} = -\frac{(Ca_2 - Ca_{2c})}{t} \cdot \frac{W_w}{W_c}$          (4)
where $A_{T1}$ and $Ca_1$ are $A_T$ and $Ca^{2+}$ concentrations at the start of the incubations (in μmol kg$^{-1}$
), $A_{T2}/A_{T2c}$ and $Ca_2/Ca_{2c}$ are $A_T$ and $Ca^{2+}$ concentrations at the end of the incubations,
respectively with and without corals, t is the incubation duration in h, $W_w$ and $W_c$ are
respectively the mass of seawater (average between initial and final weights) and the coral
skeleton dry weight (g; DW). $G_{AT}$ and $G_{Ca}$ are therefore expressed in μmol $CaCO_3$ g DW$^{-1}$ h$^{-1}$.
Error propagation was used to estimate errors:

---

[2]seacarb: seawater carbonate chemistry with R. Gattuso, J.-P., J. M. Epitalon, H. Lavigne, J. C. Orr, B. Gentili, M. Hagens, A. Hofmann, A. Proye, K. Soetaert and J. Rae, 2018. https://cran.r-project.org/package=seacarb



$$SE_{G_{AT}} = \frac{\sqrt{SE_{AT_2}^2 + SE_{AT_{2c}}^2}}{2t} \cdot \frac{W_w}{W_c}$$    (5)
$$SE_{G_{Ca}} = \frac{\sqrt{SE_{Ca_2}^2 + SE_{Ca_{2c}}^2}}{t} \cdot \frac{W_w}{W_c}$$    (6)
Coral calcification rates based on $^{45}$Ca incorporation were estimated using measured
seawater activity and activity recorded in the skeleton digest. Rates were then normalized per
g skeleton dry weight using the formula:
$$G_{45_{Ca}} = \frac{Activity_{sample} \cdot \frac{Ca}{Activity_{seawater}}}{W_c \cdot t}$$    (7)
where Activity$_{sample}$ is the average of counts per minute (CPM) of three 200 μL
aliquots from the dissolved skeleton sample, Activity$_{seawater}$ is the total CPM in the 1 mL
seawater samples, Ca is the [Ca$^{2+}$] measured in the corresponding samples (average between
initial and final values, μmol kg$^{-1}$) and further converted to μmol L$^{-1}$ considering a
temperature of 25 °C and a salinity of 38, $W_c$ is the skeleton dry weight (in g) and t the
incubation duration (h). $G_{45_{Ca}}$ is therefore expressed in μmol CaCO$_3$ g DW$^{-1}$ h$^{-1}$. The standard
errors for these calcification rate estimates were propagated based on standard errors
associated with the measurements of triplicate samples for both Activity$_{sample}$ and [Ca$^{2+}$].
The precipitation of calcium carbonate minerals (G) during the incubation interval was
also estimated using measured $\delta^{13}$C values and isotope mass balance calculations [eq. (8) and
(9) below]. The CO$_2$ released during phosphoric acid digestion is derived from two sources:
new coral CaCO$_3$ and previously accreted skeletal carbonate mineral. The new carbon
acquired in each measured nubbins ($\delta^{13}$C$_N$) was assumed to have the same carbon isotope
composition as the labelled seawater $C_T$ (average between initial and final level, $\delta^{13}$C-$C_T \sim$



1,400-1,700‰). The previously accreted skeletal material was assumed to have a $\delta^{13}C$ value
equal to the measured value for the background sample ($\delta^{13}C_P$). The $\delta^{13}C$ value ($\delta^{13}C_M$),
representing the mixture of new calcified material and previously accreted carbonate mineral,
is then calculated the following mixing equation:
$\delta^{13}C_M = f_G \cdot \delta^{13}C_N + (1 - f_G) \cdot \delta^{13}C_P$                   (8)
where $f_G$ is the fraction of the calcium carbonate mineral precipitated during the experiment,
and $\delta^{13}C_N$ and $\delta^{13}C_P$ are the carbon isotope compositions of the newly precipitated and
previously accreted calcium carbonate, respectively. Equation (8) was solved for $f_G$ to
determine the calcium carbonate precipitated during the incubation using:
$G_{13_C} = \frac{f_G}{t \cdot M_{CaCO_3}} \cdot 1e^6$                   (9)
where $M_{CaCO3}$ is the molar mass of calcium carbonate (g mol$^{-1}$) and t is the incubation
duration in h. $G_{13C}$ are therefore expressed in μmol CaCO$_3$ g DW$^{-1}$ h$^{-1}$. The standard errors for
these calcification rate estimates were calculated based on standard errors associated with the
triplicate measurements of $\delta^{13}C_P$ and $\delta^{13}C_N$.

Model-II linear regressions (Sokal and Rohlf, 1995) were used to compare net

calcification rates obtained with the different methods. All regressions were performed using
function "lmodel2" of the package lmodel2[3] on the R software.

---

[3]lmodel2: Model II Regression, Legendre P. and J. Oksanen, 2018. https://cran.r-project.org/package=lmodel2



## 3. Results


Environmental conditions at the start of the different incubations are shown in Table 2.
All incubations performed under ambient $pH_T$ (~8.05) were conducted under carbonate
chemistry favorable to calcification with saturation states with respect to aragonite ($\Omega_a$) well
above 1 (average of $4.0 \pm 0.1$ over the four incubations). In contrast, during experiments at
low $pH_T$ (initial $pH_T \sim 7.2$), seawater was corrosive with respect to aragonite ($\Omega_a \sim 0.75$).
However, as pH was not regulated during the incubations (see previous section), it increased,
at lowered pH, to an average of $7.75 \pm 0.03$ (n = 5) in dark conditions and to an average of
$7.84 \pm 0.03$ in light conditions (n = 5). Evolution of pH in control beakers (final $pH_T$ of 7.78
and 7.48; n = 1 for both in the light and in the dark, respectively) showed that the observed
increase in beakers with corals was due to the additive effects of biological control
(photosynthesis minus respiration and calcification) and exchanges at the interface in the
light, and mostly due to $CO_2$ exchange with air during the much longer incubations performed
in the dark. Assuming linear variations with time, the average conditions of the carbonate
chemistry in the lowered pH experiments were slightly favorable to aragonite production ($\Omega_a$
$= 1.4 \pm 0.2$ in the dark, n = 5 and $1.6 \pm 0.05$ in the light, n = 5). Under ambient pH conditions
(both for $^{45}$Ca and $^{13}$C incubations), pH did not change during incubations in the light
(average final $pH_T$ of $8.05 \pm 0.03$, n = 12, data not shown) while it decreased in the dark, due
to respiration and calcification, to reach an average $pH_T$ level of $7.62 \pm 0.07$, n = 12, data not
shown). In control beakers under ambient pH, $pH_T$ slightly increased in the light (8.09, n = 2)
and did not change in the dark (8.05, n = 2).
$^{45}$Ca activities in seawater did not change during the incubations, reaching a final
activity of $16.1 \pm 1.2$ (n = 12) and $28.5 \pm 0.6$ (n = 10) Bq mL$^{-1}$ under ambient and lowered pH



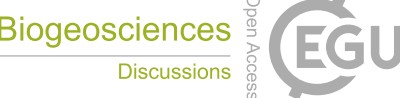

conditions, respectively (including both dark and light incubations, data not shown).
Furthermore, for all incubations, these values are similar to those measured in beakers without
corals (control, data not shown). Under ambient pH levels (no incubation at lowered pH),
seawater was enriched in $^{13}$C ($\delta^{13}$C-$C_T$) from a background level of $0.26 \pm 0.05$‰ (n = 3) to
$1{,}740 \pm 4.7$‰ (n = 2) and $1{,}634 \pm 11$‰ (n = 2) in the light and dark, respectively. During
light condition incubations, $\delta^{13}$C-$C_T$ levels decreased to an average of $1{,}636 \pm 10$‰ (n = 6,
data not shown) while they decreased to an average of $1{,}466 \pm 24$‰ in dark conditions (n = 6,
data not shown). Incubations in control beakers (without corals) showed that the majority of
$\delta^{13}$C-$C_T$ loss for both types of incubations (light and dark) was due to $^{13}$C incorporation by
corals with a minor effect of gas exchanges at the interface (data not shown).

Changes in $A_T$ and [$Ca^{2+}$] in beakers containing corals as compared to control beakers,

during all sets of incubations, are shown in Table 3. Both variables declined in all incubations
as a consequence of coral calcification. Changes in $A_T$ during incubations in control beakers
(data not shown) were comprised between 0.1 and 1.1% of the initial level. Similar results
were observed for [$Ca^{2+}$] with a relative change comprised between 0.05 and 1.15% of the
initial value. These minimal changes were corroborated with no measurable changes in
seawater weight between the start and the end of all incubations (data not shown), showing
that evaporation, if any, was minimal using our experimental set-up over the considered
incubation times. At ambient pH levels, decreases in $A_T$ and [$Ca^{2+}$] (average of $-380 \pm 97$ and
$-194 \pm 51$ µmol kg$^{-1}$ for both parameters, respectively, n = 24 including both $^{45}$Ca and $^{13}$C
incubations) were roughly similar under light and dark conditions although coral specimen
used for dark incubations were ca. 166% heavier (skeleton dry weight, see Table 1).
Incubations performed under lowered pH levels showed much lower $A_T$ and [$Ca^{2+}$] net
consumption rates than under ambient pH levels. Under these pH conditions, an extremely



high $A_T$ consumption rate was observed in one beaker (dark incubation, see Table 3) while no
changes in [$Ca^{2+}$] was observed in a total of three beakers (see Table 3). These rates have
been considered as outliers and were not included in the following analyses.

$^{45}Ca$ activities in coral skeleton reached maximum levels under ambient pH and light

conditions (average of 87.5 ± 9.1 Bq, n = 6). Although seawater was more enriched in $^{45}Ca$ at
the lower pH levels (see above), $^{45}Ca$ activity in corals incubated under these conditions were
much lower with lowest values measured in the dark (average of 19.6 ± 9.1 Bq, n = 5). $\delta^{13}C$
levels measured in coral skeletons (-3.69 to 8.92‰) showed significant enrichment as
compared to background levels (-3.97 ± 0.35‰, n = 9).

Estimated rates of calcification using the different techniques are presented in Table

A1 and are compared in Figs. 2, 3 and 4 as well as in Table 4. Rates were higher in the light
than in the dark and much lower rates were estimated at lowered pH. The rates measured by
alkalinity anomaly ($G_{AT}$) and calcium anomaly ($G_{Ca}$) techniques were highly correlated (Fig.
2; $R^2 = 0.98$, $p < 0.01$, n = 34). No significant difference was observed between rates
measured by the two methods (see Table 4 for the 95% confidence intervals of the slope and
intercept). The $^{45}Ca$ method provided also very similar rates than the two previous approaches
(Fig. 3; $G_{Ca}$ vs. $G_{45Ca}$ not shown) although the slope and the intercept of the geometric
regression between $G_{AT}$ and $G_{45Ca}$ were significantly different from 1 and 0, respectively.
Finally, the only approach that did not provide similar rates to the others was the $^{13}C$
incorporation technique. Calcification rates based on this method were systematically higher
than those measured using the other three techniques (see Table 4), and rates were not always
significantly related (e.g. $R^2 = 0.33$, $p > 0.05$, n = 12 for $G_{AT}$ vs $G_{13C}$, see Fig. 4; other
relationships not shown).

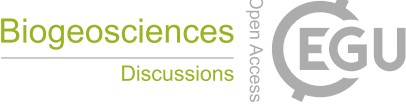

## 4. Discussion


Under all experimental conditions (ambient pH vs low pH, light vs dark), significant
consumption rates of $A_T$ and $Ca^{2+}$ as well as significant incorporation rates of $^{45}Ca$ and $^{13}C$
were observed in the zooxanthellate coral *Stylophora pistillata*. For all methods, calcification
rates were lower in dark than in light conditions. Such trends are expected as it has long been
established that calcification rates increase in zooxanthellate corals during periods in which
photosynthesis is occurring (Yonge, 1931), a process known as light-enhanced calcification
(e.g. Gattuso et al., 1999). Even under lowered pH conditions, at pH levels far below those
predicted to occur in the next decades (starting $pH_T$ of ca. 7.2, average $pH_T$ during incubations
of ca. 7.5), all corals appeared to produce calcifying structures under both light and dark
conditions. The organisms selected for this experiment were fully coated with tissues with no
exposed calcareous structures which can explain the absence of observable net dissolution
such as reported by Cohen et al. (2017) in a similar study. Since our experimental protocol
was not designed to address the potential impact of decreasing pH levels on calcification rates
of this species (no control of carbonate chemistry during incubations, no acclimation of the
organisms etc.), we will not discuss further the observed decrease of calcification rates
identified by the three techniques used at these pH levels.
Under all experimental conditions, rates of calcification calculated using the alkalinity
and the calcium anomaly techniques were highly correlated with a slope of 1 and no
significant intercept. These results are consistent with previously published data on colonies
of *Pocillopora damicornis* (Chisholm and Gattuso, 1991), *Cladocora caespitosa* (Gazeau et
al., 2015) and several other coral species (Murillo et al. 2014). Although the precision
obtained on $Ca^{2+}$ measurements is among the highest reported to date (Gazeau et al., 2015),





the alkalinity anomaly technique appears as the most appropriate to estimate calcification
rates of isolated corals (better precision, stronger signals). As observed by Murillo et al.
(2014), this is not true when an entire community including sediment is investigated. The
occurrence of several processes in the sediment that can impact $A_T$ prevents the use of this
technique. It is therefore recommended to use the calcium anomaly technique when working
in natural settings, assuming that $Ca^{2+}$ concentrations are measured with an analytical
technique as precise as the one used in our study (CV < 0.05%). Similarly, although
corrections are possible when applying the alkalinity anomaly technique on organisms that
significantly release nutrients (echinoderms, bivalves etc.), the use of the calcium anomaly
technique is highly recommended instead (Gazeau et al., 2015).

Calcification rate estimates based on changes of $A_T$ or $Ca^{2+}$ were highly correlated

with estimates based on $^{45}Ca$ incorporation in corals. These results are not consistent to those
reported by Smith and Roth (in Smith and Kinsey, 1978), Tambutté et al. (1995) and Cohen et
al. (2017). These studies revealed discrepancies between the alkalinity anomaly and the $^{45}Ca$
incorporation techniques. Smith and Roth found that rates measured with the $^{45}Ca$ method
were higher than those measured using the alkalinity anomaly technique (significant $^{45}Ca$
incorporation at $\Delta A_T = 0$). Results from both Tambutté et al. (1995) and Cohen et al. (2017)
suggested the opposite with a decrease in $A_T$ consumption without any concomitant $^{45}Ca$
incorporation. A number of reasons may explain these discrepancies. First, the present study
is the first one comparing these techniques in the same incubations, in contrast to the other
ones in which incubations for $A_T$ anomaly and $^{45}Ca$ incorporation were performed over two
consecutive days (due to radioactive contamination issues). Second, calcification expressed as
absolute changes in $A_T$ during incubations, measured during our experiment, were at least one
order of magnitude higher than measured during these studies (44,200 to 745,600 nmol vs



less than 4,000 nmol in previous experiments). Cohen et al. (2017) have shown that such
discrepancies were much higher at very low rates and that the ratio between rates estimated
based on $^{45}$Ca incorporation and $A_T$ consumption were getting closer to 1 with increasing
calcification rates. Nevertheless, even at the highest levels of calcification computed during
these studies, $^{45}$Ca-based rates were still significantly different from $\Delta A_T$-based rates, which is
in contrast with our results.

As already mentioned, although calcification rates of the present study were lower at

lowered pH levels, there was still a close to perfect agreement between the different
techniques. While the $^{45}$Ca labelling technique is thought to provide rates of gross
calcification, there is no doubt that both the $A_T$ and $Ca^{2+}$ anomaly techniques allow the
estimation of net calcification rates (gross calcification – dissolution). A full agreement of
rates computed from these methods further suggest that no dissolution of previously
precipitated $CaCO_3$ structures occurred during our study, even under lowered pH conditions.
The corals used in our experiment were fully covered with tissues which is likely the reason
that no dissolution was measured.

Furthermore, we must note that the protocol for $^{45}$Ca incorporation considered in our

study differed from the one used in the above-mentioned past studies. A much smaller activity
was used (0.025 kBq mL$^{-1}$) compared to Tambutté et al. (1995; 40 kBq mL$^{-1}$) and Cohen et al.
(2017; 9 kBq mL$^{-1}$). Moreover, in contrast to Cohen et al. (2017), rates were not corrected for
$^{45}$Ca incorporation on the skeleton of dead corals. This choice was motivated by the absence
of detectable radioactivity on bare skeletons exposed for 7 h and treated with the same
protocol than one used in our study (Lanctôt, pers. comm.).

To the best of our knowledge, this is the first study comparing calcification rates

measured using the $^{13}$C labelling technique to the more widely used alkalinity and calcium



anomaly techniques. It shows that [13]C-derived rates were systematically higher and much
more variable (with large uncertainties) than the ones estimated using the two other
techniques. As already mentioned, several studies have shown that most of the carbon
precipitated in the skeleton comes from coral and its symbiotic zooxanthellae (e.g. Erez,
1978; Furla et al., 2000), leading to an underestimation of calcification rates based on
labelled, radioactive carbon incorporation. As there is no reason for [13]C to behave differently,
our results appear inconsistent with a metabolic source of carbon. As the nubbins were treated
following the same protocol as for [45]Ca incorporation measurements, it is unclear why much
stronger [13]C incorporation were obtained and why variability is so high. Before better insights
on such discrepancies can be developed, we recommend to avoid this technique to estimate
coral calcification rates.

Although our study was designed to compare different techniques to estimate

calcification rates and not to define the best experimental approach to study the effects of
ocean acidification on coral species using these different approaches, our results provide some
insights that we further discuss in the following section. Measuring and comparing
calcification rates of organisms under varying pH conditions requires the careful choice of a
volume and a time interval such that the precision of the calcification rate measurement is
large enough to observe significant signals and that the change in carbonate chemistry
parameters between the beginning and end of the incubation is small compared to the range of
these parameters in the different treatments (Langdon et al. 2010).

Table 5 illustrates the incubation time necessary to obtain measurable changes

considering the ratio between incubation volume and coral size chosen for our study. As the
[13]C incorporation method did not provide reliable rates, this technique was not considered in
this analysis. The threshold for significant signals was set at 10-fold the analytical precision



of the instruments (Langdon et al. 2010) for $A_T$ and $Ca^{2+}$ measurements (1.2 and 2.9 μmol kg⁻
¹, respectively) and above the detection limit of 15 cpm for $^{45}Ca$ activity estimated. Maximum
incubation times have been determined considering a maximum decrease of $C_T$ by 10%
(Langdon et al. 2010).

Under light and ambient pH conditions, even if the ratio between incubation volume

and nubbin size is much higher than for previous similar studies (e.g. Cohen et al. 2017), all
methods would allow a precise estimation of calcification rates over very short incubation
times (~15 min to 1 h, depending on the method) while leading to moderate changes in $C_T$ (<
10%). In the dark, and at ambient pH conditions, in the absence of $C_T$ increase by
photosynthesis, the increase of $C_T$ due to respiration, of which only a minor portion is
compensated by calcification, narrows the possible incubation period to 12 h. However, this is
still much larger than the incubation time allowing to obtain a significant signal with the three
methods (~20 min to ~2 h). At lower pH, both under light and dark conditions, and using
open systems without a continuous pH regulation as in our study, it is obvious that the
calcium anomaly technique is not well adapted to this experimental protocol. Indeed, as a
consequence of lower calcification rates at lower pH and important $CO_2$ degassing, incubation
times necessary to obtain significant signals using this technique are too large to maintain the
carbonate parameters within an acceptable range ($\Delta C_T < 10\%$). This is not insurmountable as
a continuous regulation of pH using for instance pure $CO_2$ bubbling or incubations performed
in a closed container (i.e. without contact to the atmosphere) would alleviate these problems.
Nevertheless, if such experimental protocols cannot be followed, our results show that the
alkalinity anomaly and $^{45}Ca$ incorporation techniques are still sensitive enough, at lowered
pH, to estimate reliable calcification rates in zooxanthellate corals maintained in open-



systems without continuous pH regulation, while maintaining acceptable changes in the
carbonate chemistry.
In conclusion, the present study is the first one allowing a direct (i.e. during the same
incubations) comparison of three methods used to estimate coral calcification rates, the
calcium and alkalinity anomaly techniques and the $^{45}$Ca incorporation technique. These
methods provided very consistent calcification rates of the coral *Stylophora pistillata*
independently of the conditions set for the incubations (light vs dark, ambient vs low pH).
Among these three methods, the alkalinity anomaly and the $^{45}$Ca incorporation techniques
appear to be the most sensitive allowing the quantification of coral calcification rates without
significant changes in targeted environmental conditions. In contrast, the $^{13}$C incorporation
technique did not provide reliable calcification rates and its use is not recommended until
further investigations clarify the discrepancies. Finally, this study was restricted to a single
coral species and used nubbins fully covered with tissues. Conducting similar comparison
studies with other coral species as well as other major calcifying groups widely studied in the
context of ocean acidification (e.g. coralline algae, molluscs etc…) would be necessary.



## Acknowledgements

This work was supported by the IAEA's Ocean Acidification International Coordination

Center (OA-ICC) and the IAEA-ICTP Sandwich Training Educational Programme (STEP)

and Project "Strengthening the National System for Analysis of the Risks and Vulnerability of

Cuban Coastal Zone Through the Application of Nuclear and Isotopic Techniques" National

Program PNUOLU /4-1/ 2 No. /2017 of the National Nuclear Agency (AENTA). We thank

the Monaco government, the Centre Scientifique de Monaco for propagating and maintaining

the coral nubbins and Samir Alliouane for technical assistance for total alkalinity and calcium

measurements.



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





Table 1. Experimental details for the series of incubations of the coral *Stylophora pistillata* performed under ambient and low pH, and in
the light and dark following $^{45}$Ca or $^{13}$C labelling. The ratio $W_w$:$W_c$ corresponds to the ratio between seawater weight (g) and skeletal dry
weight (g). Values represent mean ± standard deviation (SD); n is the number of true replicates considered for each experiment.

| pH conditions | Ambient (n = 6) | | | | | | Lowered (n = 5) | |
|---|---|---|---|---|---|---|---|---|
| Added label | $^{45}$Ca | | | | $^{13}$C | | $^{45}$Ca | |
| Light conditions | Light | Dark | | Light | Dark | | Light | Dark |
| Coral size (mm) | 33.2 ± 1.5 | 44.7 ± 1.5 | | 36.3 ± 2.2 | 50.2 ± 1.7 | | 26.0 ± 1.6 | 28.9 ± 1.9 |
| Coral Skeleton dry weight (g) | 2.5 ± 0.5 | 3.8 ± 0.7 | | 2.6 ± 0.5 | 4.7 ± 0.5 | | 2.1 ± 0.2 | 2.8 ± 0.4 |
| Ratio $W_w$:$W_c$ | 126.4 ± 25.6 | 81.9 ± 14.7 | | 106.9 ± 24.5 | 67.8 ± 7.5 | | 146.5 ± 14.3 | 110.0 ± 12.4 |
| Incubation time (h) | 8 | 8 | | 9.12 | 9.12 | | 5 | 11 |

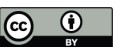


Table 2. Environmental conditions at the start of incubations of the coral *Stylophora pistillata*. pH on the total scale ($pH_T$), partial pressure
of $CO_2$ ($pCO_2$ in µatm), total alkalinity ($A_T$ in µmol kg$^{-1}$), dissolved inorganic carbon ($C_T$ in µmol kg$^{-1}$), saturation states with respect to
aragonite ($\Omega_a$) and calcite ($\Omega_c$) as well as calcium concentrations ([Ca$^{2+}$] in µmol kg$^{-1}$) are presented. Labelled seawater $^{45}$Ca activity
(Activity$_{seawater}$ in Bq mL$^{-1}$) and the isotopic level, after enrichment, of the seawater dissolved inorganic carbon pool ($\delta^{13}$C-$C_T$ in ‰) are
also shown. Means ± standard deviation of analytical triplicates (duplicates for $\delta^{13}$C-$C_T$) are shown when available.

| pH conditions | Ambient | | | | Lowered | |
| --- | --- | --- | --- | --- | --- | --- |
| Added label | $^{45}$Ca | | $^{13}$C | | $^{45}$Ca | |
| Light conditions | Light | Dark | Light | Dark | Light | Dark |
| $pH_T$ | 8.05 | 8.05 | 8.06 | 8.05 | 7.21 | 7.24 |
| $pCO_2$ | 427.6 ± 8.2 | 438.8 ± 8.5 | 425.6 ± 8.2 | 424.1 ± 8.2 | 3,727.2 ± 66.8 | 3,460.1 ± 62.1 |
| $A_T$ | 2,556.0 ± 0.5 | 2,620.0 ± 0.7 | 2,615.2 ± 0.6 | 2,535.9 ± 1.8 | 2,558.4 ± 0.3 | 2,552.9 ± 2.4 |
| $C_T$ | 2,206.4 ± 7.4 | 2,264.1 ± 7.6 | 2,252.9 ± 7.7 | 2,188.2 ± 7.6 | 2,597.1 ± 2.5 | 2,579.8 ± 3.5 |
| $\Omega_a$ | 3.9 ± 0.2 | 4.0 ± 0.2 | 4.1 ± 0.2 | 3.9 ± 0.2 | 0.7 ± 0.0 | 0.8 ± 0.0 |
| $\Omega_c$ | 5.9 ± 0.3 | 6.1 ± 0.3 | 6.2 ± 0.3 | 5.9 ± 0.3 | 1.1 ± 0.1 | 1.2 ± 0.1 |
| [Ca$^{2+}$] | 11,179.6 ± 0.0 | 11,164.0 ± 2.0 | 11,096.5 ± 13.4 | 11,098.5 ± 2.8 | 11,281.2 ± 5.5 | 11,277.6 ± 0.3 |
| Activity$_{seawater}$ | 16.6 | 15.1 | - | - | 28.5 | 30.4 |
| $\delta^{13}$C-$C_T$ | - | - | 1,740 ± 4.7 | 1,634 ± 11 | - | - |





Table 3. Changes in total alkalinity ($A_T$) and calcium concentrations ($[Ca^{2+}]$) during the different types of incubations as compared to control beakers: $\Delta A_T = A_{T2} - A_{T2c}$, $\Delta[Ca^{2+}] = Ca_2 - Ca_{2c}$, both expressed in µmol kg⁻¹. Standard errors (SE) have been calculated as $\sqrt{SE^2_{AT_2} + SE^2_{AT_{2c}}}$ and $\sqrt{SE^2_{Ca_2} + SE^2_{Ca_{2c}}}$ for $A_T$ and $[Ca^{2+}]$, respectively, where SE correspond to standard errors associated with the measurement of three analytical replicates per sample. $^{45}$Ca activity (Activity$_{sample}$ in Bq) and $^{13}$C incorporation ($\delta^{13}C_M$ in ‰) of sampled corals are also shown. Values of $^{45}$Ca activity and $\delta^{13}$C are mean ± standard error of the mean (SE) associated with the measurement of three aliquots for each coral.

| Experiment | Beaker# | $\Delta A_T$ | SE $\Delta A_T$ | $\Delta[Ca^{2+}]$ | SE $\Delta[Ca^{2+}]$ | Activity$_{sample}$ | SEActivity$_{sample}$ | $\delta^{13}C_M$ | SE $\delta^{13}C_M$ |
|---|---|---|---|---|---|---|---|---|---|
| Ambient pH - $^{45}$Ca - Light | 1 | -343.6 | 1.3 | -166.0 | 6.0 | 78.5 | 1.9 | - | - |
|  | 2 | -368.9 | 0.9 | -174.1 | 5.1 | 86.5 | 2.9 | - | - |
|  | 3 | -336.9 | 0.9 | -181.3 | 2.7 | 78.2 | 2.3 | - | - |
|  | 4 | -364.3 | 0.9 | -190.6 | 6.3 | 85.2 | 0.8 | - | - |
|  | 5 | -406.7 | 0.7 | -225.6 | 1.4 | 95.7 | 2.6 | - | - |
|  | 6 | -407.5 | 1.2 | -175.9 | 1.1 | 100.6 | 3.5 | - | - |
| Ambient pH - $^{13}$C - Light | 1 | -386.3 | 1.5 | -195.0 | 3.8 | - | - | -1.4 | 2.0 |
|  | 2 | -422.6 | 1.3 | -206.8 | 4.2 | - | - | 1.8 | 3.2 |




|  |  |  |  |  |  |  |  |  |
|---|---|---|---|---|---|---|---|---|
| 3 | -405.4 | 1.9 | -200.9 | 2.1 | - | - | 3.4 | 5.1 |
| 4 | -481.6 | 1.3 | -253.2 | 2.0 | - | - | 1.1 | 2.0 |
| 5 | -498.4 | 1.3 | -260.5 | 5.7 | - | - | 0.8 | 0.7 |
| 6 | -618.1 | 1.8 | -317.7 | 4.4 | - | - | 0.1 | 1.8 |
| Ambient pH - $^{13}$C - Dark 1 | -300.5 | 1.4 | -168.9 | 0.6 | - | - | -0.3 | 1.3 |
| 2 | -440.8 | 1.4 | -220.7 | 2.5 | - | - | -3.0 | 0.5 |
| 3 | -223.5 | 1.9 | -135.1 | 0.8 | - | - | -3.1 | 0.6 |
| 4 | -347.3 | 1.1 | -185.3 | 0.2 | - | - | 0.5 | 5.4 |
| 5 | -571.7 | 1.3 | -301.7 | 1.2 | - | - | 0.6 | 2.1 |
| 6 | -434.5 | 1.3 | -224.6 | 3.7 | - | - | 0.7 | 6.1 |
| Ambient pH - $^{45}$Ca - Dark 1 | -290.2 | 1.6 | -157.9 | 2.2 | 56.44 | 1.24 | - | - |
| 2 | -274.3 | 1.2 | -130.4 | 4.4 | 50.1 | 0.74 | - | - |
| 3 | -300.8 | 1.3 | -168.3 | 0.9 | 57.17 | 1.75 | - | - |
| 4 | -327.0 | 2.7 | -139.3 | 5.3 | 66.24 | 0.69 | - | - |
| 5 | -342.8 | 1.2 | -172.6 | 3.0 | 68.37 | 3.11 | - | - |
| 6 | -228.3 | 1.8 | -113.4 | 2.5 | 52.36 | 2.49 | - | - |





| | | | | | | | | |
|---|---|---|---|---|---|---|---|---|
| Lowered pH - $^{45}$Ca - Light | 1 | -59.3 | 2.2 | -1.6* | 6.9 | 20.2 | 0.7 | - | - |
| | 2 | -44.2 | 2.2 | -11.0 | 2.2 | 15.3 | 0.4 | - | - |
| | 3 | -71.3 | 2.8 | -28.0 | 5.9 | 22.5 | 0.3 | - | - |
| | 4 | -70.2 | 2.4 | -35.7 | 7.6 | 23.4 | 0.4 | - | - |
| | 5 | -56.4 | 2.5 | -19.6 | 7.1 | 20 | 0.9 | - | - |
| Lowered pH - $^{45}$Ca - Dark | 1 | -745.6* | 13.2 | 0.8* | 0.3 | 14.5 | 0.2 | - | - |
| | 2 | -52.4 | 2.1 | -1.0* | 1.0 | 22.1 | 0.3 | - | - |
| | 3 | -50.5 | 2.1 | -22.5 | 2.8 | 22.1 | 0.1 | - | - |
| | 4 | -54.3 | 2.1 | -30.3 | 8.5 | 23.3 | 0.4 | - | - |
| | 5 | -99.4 | 2.1 | -32.8 | 4.1 | 16.1 | 0.1 | - | - |





Table 4. Model-II regression results of the comparison between calcification rates estimated
using the different methods considered in this study: the alkalinity and calcium anomaly
techniques ($G_{AT}$ and $G_{Ca}$, respectively) as well as the $^{45}Ca$ and $^{13}C$ incorporation techniques
($G_{45Ca}$ and $G_{13C}$, respectively). The number of samples (n), the regression coefficient ($R^2$), as
well as the slope and intercept (including their 95% confidence intervals, 95% CI) are shown
for each comparison. Few identified outliers (n = 4) have been removed from the analyses,
see Table 3 and Table A1.

| Methods compared | n | $R^2$ | Slope | | | Intercept | | |
|---|---|---|---|---|---|---|---|---|
| | | | Value | 95% CI | | Value | 95% CI | |
| | | | | Low | High | | Low | High |
| $G_{AT}$ *vs.* $G_{Ca}$ | 32 | 0.98 | 0.95 | 0.90 | 1.00 | 0.09 | 0.00 | 0.18 |
| $G_{AT}$ *vs.* $G_{45Ca}$ | 21 | 0.99 | 0.94 | 0.90 | 0.98 | 0.09 | 0.03 | 0.15 |
| $G_{Ca}$ *vs.* $G_{45Ca}$ | 20 | 0.97 | 1.00 | 0.91 | 1.09 | -0.06 | -0.20 | 0.07 |
| $G_{AT}$ *vs.* $G_{13C}$ | 12 | 0.33 | 0.49 | 0.05 | 1.2 | 0.77 | -1.2 | 2.1 |
| $G_{Ca}$ *vs.* $G_{13C}$ | 12 | 0.32 | 0.46 | 0.03 | 1.1 | 0.94 | -0.9 | 2.2 |






Table 5. Incubation times ($t_{min}$; h) necessary to obtain significant signals using the three
methods: the alkalinity anomaly technique ($A_T$), the calcium anomaly technique ($Ca^{2+}$) and
the $^{45}Ca$ incorporation techniques ($^{45}Ca$), see text for calculation procedures. $t_{max}$ (h) is the
maximum incubation time to maintain carbonate chemistry within an acceptable range ($\Delta C_T <$
10%). The ratios between incubation volume (in mL) and the size of the nubbins (in cm),
considered in our study for the different sets of incubations (Ambient pH vs Low pH; Light vs
Dark), are also shown. $t_{min}$ values are noted in bold when higher than $t_{max}$.

|  | Ratio V:S | $t_{min}$ (h) | | | $t_{max}$ (h) |
|---|---|---|---|---|---|
|  |  | $A_T$ | $Ca^{2+}$ | $^{45}Ca$ |  |
| Ambient pH – Light | 77-95 | 0.26 | 1.00 | 0.6 | 49.7 |
| Ambient pH – Dark | 59-69 | 0.33 | 2.10 | 1.5 | 12.1 |
| Lowered pH – Light | 109-121 | 1.25 | **6.15** | 1.1 | 2.8 |
| Lowered pH – Dark | 95-109 | 1.60 | **11.20** | 3.4 | 4.4 |






Table A1. Calcification rates estimated by the different methods considered in this study: the alkalinity and calcium anomaly techniques ($G_{AT}$ and $G_{Ca}$, respectively) as well as the $^{45}Ca$ and $^{13}C$ incorporation techniques ($G_{45Ca}$ and $G_{13C}$, respectively). All rates are mean ± standard errors of the mean (SE) and are expressed in μmol CaCO$_3$ g DW$^{-1}$ h$^{-1}$.

| Experiment | Beaker# | $G_{AT}$ | SE $G_{AT}$ | $G_{Ca}$ | SE $G_{Ca}$ | $G_{45Ca}$ | SE $G_{45Ca}$ | $G_{13C}$ | SE $G_{13C}$ |
|---|---|---|---|---|---|---|---|---|---|
| Ambient pH - $^{45}Ca$ - Light | 1 | 3.28 | 0.01 | 3.17 | 0.11 | 3.41 | 0.08 | NA | NA |
| | 2 | 3.21 | 0.01 | 3.03 | 0.09 | 3.29 | 0.11 | NA | NA |
| | 3 | 2.69 | 0.01 | 2.89 | 0.04 | 2.77 | 0.08 | NA | NA |
| | 4 | 3.38 | 0.01 | 3.54 | 0.12 | 3.48 | 0.03 | NA | NA |
| | 5 | 2.41 | 0.00 | 2.68 | 0.02 | 2.53 | 0.07 | NA | NA |
| | 6 | 2.43 | 0.01 | 2.10 | 0.01 | 2.65 | 0.09 | NA | NA |
| Ambient pH - $^{13}C$ - Light | 1 | 3.26 | 0.01 | 3.29 | 0.06 | NA | NA | 1.92 | 1.35 |




|  |  |  |  |  |  |  |  |  |
|---|---|---|---|---|---|---|---|---|
| 2 | 3.30 | 0.01 | 3.23 | 0.07 | NA | NA | 4.27 | 2.27 |
| 3 | 3.09 | 0.01 | 3.06 | 0.03 | NA | NA | 5.47 | 3.66 |
| 4 | 2.98 | 0.01 | 3.14 | 0.02 | NA | NA | 3.74 | 1.36 |
| 5 | 2.80 | 0.01 | 2.92 | 0.06 | NA | NA | 3.49 | 0.41 |
| 6 | 2.73 | 0.01 | 2.81 | 0.04 | NA | NA | 3.00 | 1.22 |
| Ambient pH - $^{13}$C - Dark 1 | 1.33 | 0.01 | 1.50 | 0.01 | NA | NA | 2.58 | 0.79 |
| 2 | 1.63 | 0.01 | 1.63 | 0.02 | NA | NA | 0.68 | 0.23 |
| 3 | 0.85 | 0.01 | 1.03 | 0.01 | NA | NA | 0.61 | 0.30 |
| 4 | 1.24 | 0.00 | 1.32 | 0.00 | NA | NA | 3.14 | 3.67 |
| 5 | 1.96 | 0.00 | 2.07 | 0.01 | NA | NA | 3.21 | 1.35 |
| 6 | 1.42 | 0.00 | 1.46 | 0.02 | NA | NA | 3.28 | 4.16 |



| | | | | | | | | |
|---|---|---|---|---|---|---|---|---|
| Ambient pH - $^{45}$Ca - Dark | 1 | 1.59 | 0.01 | 1.72 | 0.02 | 1.54 | 0.03 | NA | NA |
| | 2 | 1.39 | 0.01 | 1.32 | 0.04 | 1.26 | 0.02 | NA | NA |
| | 3 | 1.46 | 0.01 | 1.64 | 0.01 | 1.43 | 0.04 | NA | NA |
| | 4 | 1.29 | 0.01 | 1.10 | 0.04 | 1.33 | 0.01 | NA | NA |
| | 5 | 1.44 | 0.01 | 1.45 | 0.03 | 1.44 | 0.07 | NA | NA |
| | 6 | 0.75 | 0.01 | 0.75 | 0.02 | 0.89 | 0.04 | NA | NA |
| Lowered pH - $^{45}$Ca - Light | 1 | 1.00 | 0.04 | 0.05* | 0.23 | 0.85 | 0.03 | NA | NA |
| | 2 | 0.66 | 0.03 | 0.33 | 0.07 | 0.58 | 0.02 | NA | NA |
| | 3 | 0.96 | 0.04 | 0.75 | 0.16 | 0.80 | 0.01 | NA | NA |
| | 4 | 1.04 | 0.04 | 1.06 | 0.23 | 0.94 | 0.02 | NA | NA |
| | 5 | 0.75 | 0.03 | 0.52 | 0.19 | 0.73 | 0.03 | NA | NA |





| Lowered pH - $^{45}Ca$ - Dark | | | | | | | | |
|---|---|---|---|---|---|---|---|---|
| 1 | 4.05* | 0.07 | -0.01* | 0.00 | 0.20 | 0.00 | NA | NA |
| 2 | 0.22 | 0.01 | 0.01* | 0.01 | 0.24 | 0.00 | NA | NA |
| 3 | 0.25 | 0.01 | 0.22 | 0.03 | 0.30 | 0.00 | NA | NA |
| 4 | 0.30 | 0.01 | 0.34 | 0.10 | 0.35 | 0.01 | NA | NA |
| 5 | 0.48 | 0.01 | 0.32 | 0.04 | 0.21 | 0.00 | NA | NA |

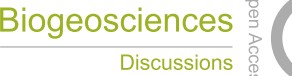

Figure captions
Fig. 1. Scheme of the polyethylene container in which a coral nubbin is suspended with a
nylon line and covered with a transparent film.
Fig. 2. Calcification rates estimated based on the alkalinity anomaly technique ($G_{AT}$) as a
function of calcification rates estimated based on the calcium anomaly technique ($G_{Ca}$). The
dashed line represents the 1:1 relationship while the full line represents the model-II
regression relationship. Horizontal error bars represent standard errors (SE) associated with
the estimation of $G_{Ca}$. Vertical error bars representing SE associated with the estimation of
$G_{AT}$ are too small to be visible. The corresponding dataset can be found in Table A1.
Fig. 3. Calcification rates estimated based on the alkalinity anomaly technique ($G_{AT}$) as a
function of calcification rates estimated based on the $^{45}Ca$ incorporation technique ($G_{45Ca}$).
The dashed line represents the 1:1 relationship while the full line represents the model-II
regression relationship. Horizontal error bars represent standard errors (SE) associated with
the estimation of $G_{45Ca}$. Vertical error bars representing SE associated with the estimation of
$G_{AT}$ are too small to be visible. The corresponding dataset can be found in Table A1.
Fig. 4. Calcification rates estimated based on the alkalinity anomaly technique ($G_{AT}$) as a
function of calcification rates estimated based on $^{13}C$ incorporation technique ($G_{13C}$). The
dashed line represents the 1:1 relationship while the full line represents the model-II
regression relationship. Horizontal error bars represent standard errors (SE) associated with
the estimation of $G_{13C}$. Vertical error bars representing SE associated with the estimation of
$G_{AT}$ are too small to be visible. The corresponding dataset can be found in Table A1.



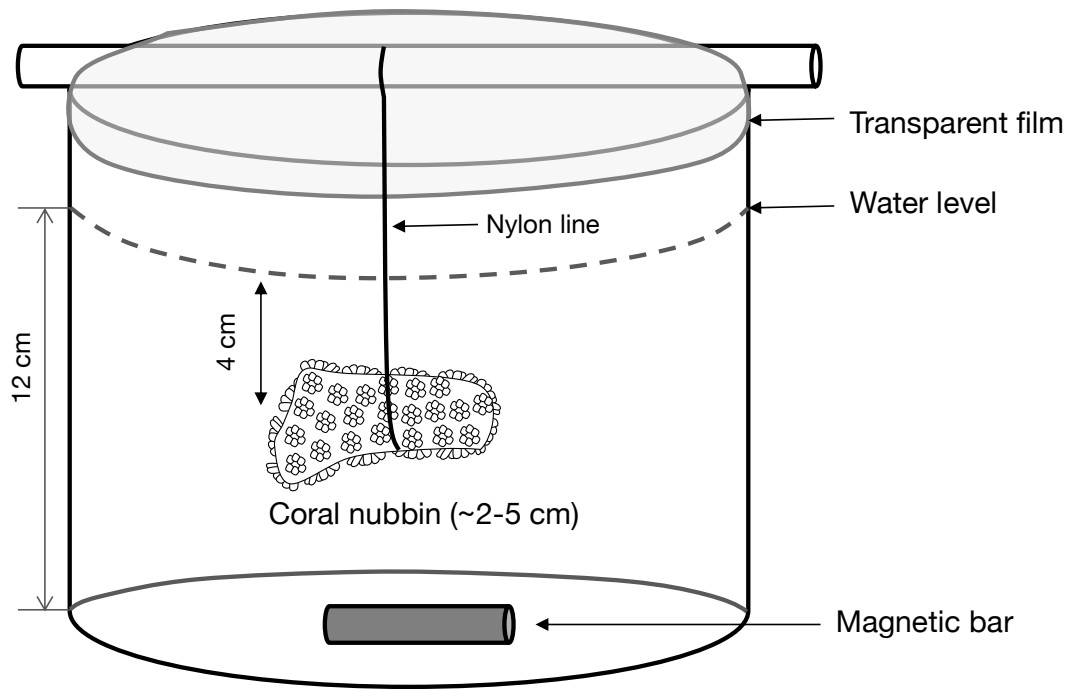


Fig. 1.



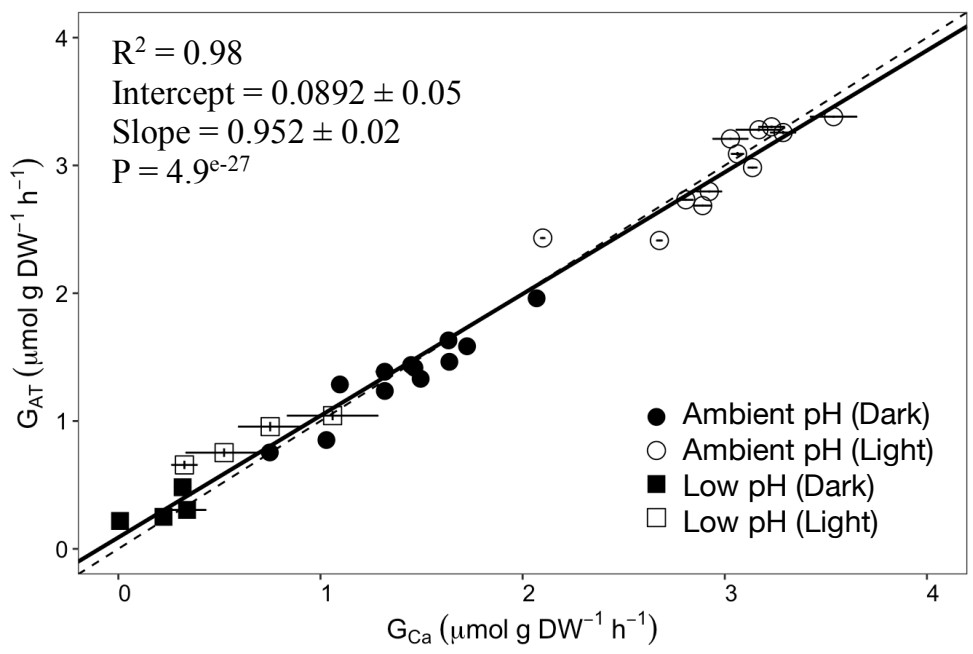


Fig. 2.





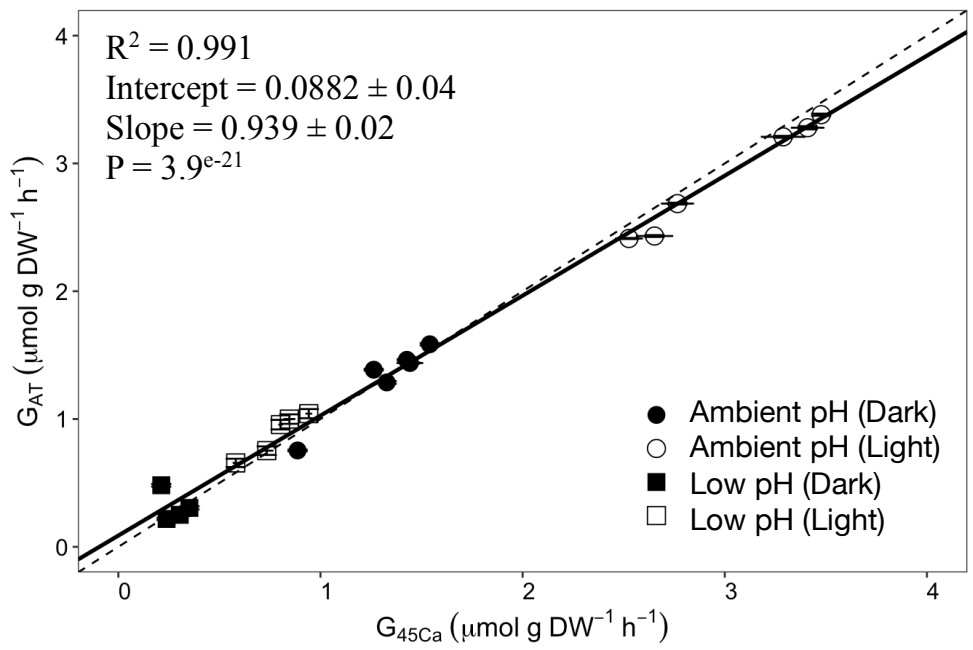


Fig. 3.



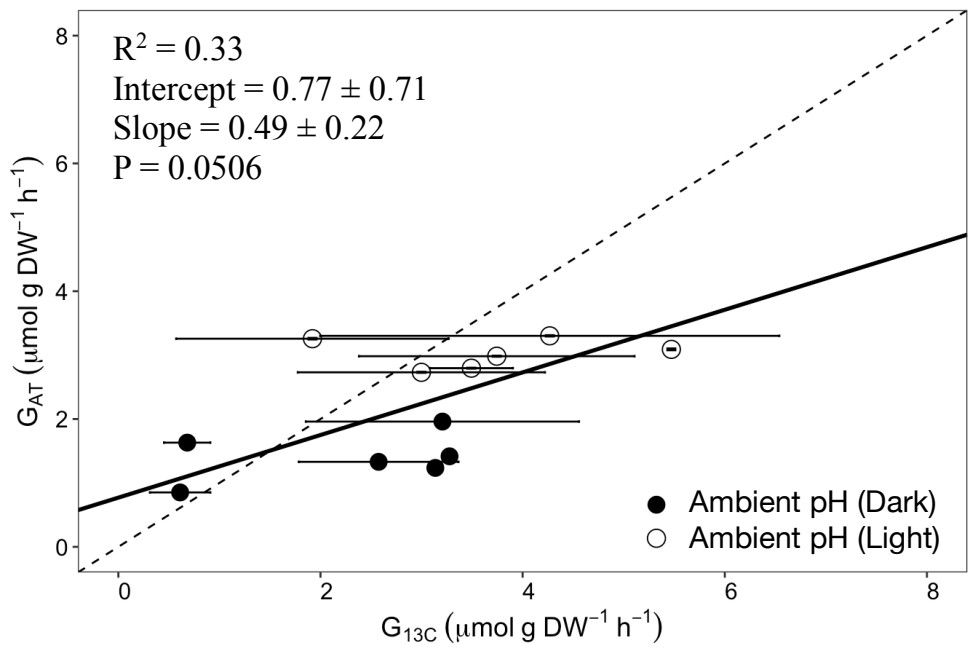


Fig. 4.