# Peer review of "Intercomparison of four methods to estimate coral calcification under various"

_Biogeosciences, 2019_

## Referee Comment (RC1) · Anonymous Referee #1 · 11 Nov 2019

This is an interesting study that compares 4 different methods for quantifying calcification rates under high and low pH conditions. The authors conclude that that alkalinity anomaly, Ca anomaly, and Ca45 methods are all in close agreement, but the 13C method is not. This is a helpful study for researchers that are trying to calculate calcification rates of individual corals. The methods are rigorous. However, I personally have only done the TA anomaly technique so hopefully the other reviewers have hands-on experience with the other 3 methods. My comments below are minor. I believe this will make a nice contribution to the coral biogeochemistry literature.

Abstract:

[Figure]

Line 27: add a comma after calcification

Line 41: This is a bit of a meta comment, but what if the 13C method is accurate and the other 3 are highly correlated, but wrong. How do we know which of these methods are "true" net calcification?

Introduction:

Line 77: You can account for changes in nutrients (by measuring nitrate, phosphate, and ammonium and incorporating into the delta TA) as well as evaporation (normalize to salinity) in the alkalinity anomaly technique.

Line 96: Replace comma with semi-colon and add comma after "therefore".

Line 113 – 114: Incorporate this sentence into the last paragraph

Methods: Line 147: replace "a" with "and"

Line 180: remove "a"

Line 265 states that initial levels are not necessary to compute calcification and only final values with and without corals are used, but line 269 says that T1 are concentrations are the start of the incubations. This is a bit confusing. Please clarify.

Line 275 – 276: Please explain the parameters in the equations.

Line 280: There is an empty box on the equation. I think it is worth discussing why different incubation times were used. Why not do them all at the same time to reduce error with changing carbonate chemistry in the background (i.e. the longest time needed to get a result from all 4 methods)?

Please add incubation temperatures to table 1 or 2

Results section throughout: Instead of saying X and Y are presented in Figures 1 and 2, make a statement about the result and cite the figure and table after. (For example, see like 368).

---

## Referee Comment (RC2) · Anonymous Referee #2 · 14 Nov 2019

This is a nice study comparing 4 different methods to measure short-term calcification rates in corals. The comparison of three less commonly used methods (calcium anomaly, 45Ca, 13C) with the commonly used alkalinity anomaly technique adds to the existing literature of method comparisons for estimating coral calcification. Furthermore, the current study has the benefit that the different methods were measured during the same incubation, minimizing the risk of other factors confounding the results. The authors show that two of the three methods are highly correlated and not significantly different from the alkalinity anomaly technique, and further provide useful recommendations on minimum and maximum incubation times for various volume to biomass ratios and techniques. Overall, this will be a useful addition to the existing

literature on coral calcification methods. As a note of caution, I do not have experience with the calcium anomaly, 45Ca and 13C methods, therefore I cannot judge the experimental protocol used for these methods.

I only have one concern regarding the data: since there was no pH control during the incubations and some incubation times were rather long, especially when conducted in the dark, significant changes in carbonate chemistry did occur over the course of these incubations. For example, pH decreased from 8.05 to 7.62 under ambient conditions in the dark due to respiration and calcification. While this is clearly stated in the Results, the Discussion on acceptable changes in carbonate chemistry largely focuses on changes in delta CT rather than pH but I don't think such a change is acceptable in studies that actually aim to detect the impacts of low pH on coral calcification. Similarly, (Riebesell et al. 2010) also recommend that changes in AT during incubations should be within 10% of starting AT, yet changes in this study were typically larger than this, except under low pH. Furthermore, there is no discussion whatsoever regarding changes in dissolved oxygen and this was also not measured, despite hypoxia/hyperoxia potentially stressing the corals. Again, while this may be less relevant for a method comparison, it is certainly relevant when making recommendations for general incubation times. I would therefore encourage the authors to discuss these aspects in more detail in the Discussion.

Specific Comments

Abstract

L32: please state the respective pH values instead of ambient and low

Introduction

L61: please also cite here other studies that recently compared various calcification methods, such as (Gazeau et al. 2015), (Schoepf et al. 2016) and (Cohen et al. 2017)

L84: "solid agreement" – this is rather colloquial and should be rephrased, e.g. "good

agreement"

L114: you could add here that this was done under different pH and light conditions

Methods

L124-138: Please provide more information on how water motion/flow was provided in the aquaria, how big the tanks were, rate of seawater renewal etc

L127: please provide more information on how many branches from how many different parent colonies were collected for each experiment

L130: what was the concentration of Artemia fed during experiment 1? This info is only provided for experiment 2

L137: please change to "biometrics parameters of the biological material"

L146: looking at Fig. 1, I wonder whether the rod to which the nylon line was attached shaded the coral from light coming from above?

L147: should be "and low pH"

L273: a description of how coral skeletal dry weight was measured is missing from the Methods. Please add.

L309: It's good to see that model II regressions were used for the analyses.

Results

L313: Table 2: why was the seawater activity much higher in experiment 2 than 1?

L316: please state whether this is SD or SE

L328: was this change in pH during incubation similar for the different methods?

L336: should be "were similar"

L361: there are also some other data with asterisks in Table 3 – I assume they are also

outliers but this is not explicitly discussed. Please clarify.

Discussion

L443: please replace "that" with "why"

L461: should be "was" x2

L492: should be "importantly"

L514: would be necessary for what? Please add.

Figures and Tables

Table 3 is very long. I think this information could be better represented in a figure showing both the average of all six replicates per treatment/method and the individual data points spread around the average.

Also, the legend does not currently explain what the asterisk next to some data means. Please add.

Table 4: please add the p-value for the regressions to the table.

References

Cohen S, Krueger T, Fine M (2017) Measuring coral calcification under ocean acidification: methodological considerations for the 45Ca-uptake and total alkalinity anomaly technique. PeerJ 5:e3749

Gazeau F, Urbini L, Cox TE, Alliouane S, Gattuso JP (2015) Comparison of the alkalinity and calcium anomaly techniques to estimate rates of net calcification. Mar Ecol Prog Ser 527:1–12

Riebesell U, Fabry VJ, Hansson L, Gattuso JP (2010) Guide to best practices for ocean acidification research and data reporting. European Comission Publications Office,

Schoepf V, Hu X, Holcomb M, Cai W-J, Li Q, Wang Y, Xu H, Warner ME, Melman

[Figure]

TF, Hoadley KD, Pettay DT, Matsui Y, Baumann JH, Grottoli AG (2016) Coral calcification under environmental change: a direct comparison of the alkalinity anomaly and buoyant weight techniques. Coral Reefs 1–13

---

## Author Comment (AC1) · 19 Dec 2019

We thank the reviewer for her/his comments and suggestions on our manuscript. We agree with most comments and modified/updated the manuscript accordingly. Below is a point-by-point reply.

This is an interesting study that compares 4 different methods for quantifying calcification rates under high and low pH conditions. The authors conclude that that alkalinity anomaly, Ca anomaly, and 45Ca methods are all in close agreement, but the 13C method is not. This is a helpful study for researchers that are trying to calculate calcification rates of individual corals. The methods are rigorous. However, I personally have

[Figure]

only done the TA anomaly technique so hopefully the other reviewers have hands-on experience with the other 3 methods. My comments below are minor. I believe this will make a nice contribution to the coral biogeochemistry literature.

Abstract

Line 27: add a comma after calcification

Done

Line 41: This is a bit of a meta comment, but what if the 13C method is accurate and the other 3 are highly correlated, but wrong. How do we know which of these methods are "true" net calcification?

Interesting comment. The reason why we reject the 13C method (as applied in our study) is not only because 13C based rates are not correlated to the other methods but also because calcification rates based on this technique are much higher and much more variable than rates based on the other methods. As mentioned in the text, it is very unlikely that dissolution was a significant process during our incubations as nubbins were fully covered with tissue, therefore there is no distinction between net and gross calcification. Now, calcification (net or gross) consumes 1 mole of carbon and 1 mole of calcium to produce 1 mole of calcium carbonate. The fact that D[Ca] and D[AT] and highly corelated following a 1:2 ratio fully confirms this. We should therefore have a 1:1 ratio between C and Ca fluxes, the fact that higher rates were obtained with the 13C technique is problematic. Finally, several studies have shown that most of the calcium used by the calcification process comes from seawater, a significant proportion of the carbon used comes from the metabolism of the organism, suggesting that rates based on C incorporation (14C or 13C) must significantly underestimate true net calcification.

Introduction

Line 77: You can account for changes in nutrients (by measuring nitrate, phosphate,

and ammonium and incorporating into the delta TA) as well as evaporation (normalize to salinity) in the alkalinity anomaly technique.

The reviewer is correct. We have added this small paragraph to deal with this comment: "This method assumes, however, that calcification is the only biological process influencing AT (Smith and Key, 1975). Nitrogen assimilation through photosynthetic activities, nitrification as well as aerobic and anaerobic remineralization of organic matter are known to impact AT through the consumption or release of nutrients (ammonium, nitrate and phosphate) and protons (Wolf-Gladrow et al. 2007). While for some group of species (e.g. bivalves, sea urchins), corrections appear necessary to take into account the effect of nutrient release on AT, changes in nutrient concentrations during incubations of isolated corals are too low (i.e. several orders of magnitude lower than changes in AT) to introduce a significant bias in the calculations (Gazeau et al. 2015)."

Furthermore, ammonium concentrations have been measured at the start and end of selected incubations (only at ambient pH) that confirmed this assumption (D[NH4] were at least 2 orders of magnitude lower than DAT).

We do not discuss here the need to correct for evaporation as this is discussed in details later in the text.

Line 96: Replace comma with semi-colon and add comma after "therefore".

Done

Line 113 – 114: Incorporate this sentence into the last paragraph

Done

Methods

Line 147: replace "a" with "and"

Done

Line 180: remove "a"

Done

Line 265 states that initial levels are not necessary to compute calcification and only final values with and without corals are used, but line 269 says that T1 are concentrations are the start of the incubations. This is a bit confusing. Please clarify.

Equations 3 and 4 present the calculation procedure showing that initial levels are not necessary to compute calcification rates as stated in the text above the equations. We believe it is important to detail these equations and do not believe this is confusing as presented. However, to make sure there is no misunderstanding we added: "where AT1 and Ca1 are AT and Ca2+ concentrations at the start of the incubations (in $\mu$mol kg-1; not used in the computations), ..."

Line 275 – 276: Please explain the parameters in the equations.

Done.

Line 280: There is an empty box on the equation.

Corrected.

I think it is worth discussing why different incubation times were used. Why not do them all at the same time to reduce error with changing carbonate chemistry in the background (i.e. the longest time needed to get a result from all 4 methods)?

We did not have this information before starting this study. Incubation times have been chosen based on practical aspects (access to the lab etc...). The fact that they differ between different incubations is not in conflict with our objective which was to compare changes in various parameters during the same incubation, not to compare different incubations between each other. A sentence has been added in the Material and Method section: "Incubation times were not fixed based on scientific considerations and differed between the different incubations due to practical constrains (i.e. access

to the lab etc. . .)."

Please add incubation temperatures to table 1 or 2

As temperature was maintained constant and at the same level for all incubations, the temperature level is now mentioned in the legend of both tables.

Results section throughout: Instead of saying X and Y are presented in Figures 1 and 2, make a statement about the result and cite the figure and table after. (For example, see like 368).

Modified accordingly.

―――――――――――――――――――――

---

## Author Response (AR2)

*We thank the reviewer for her/his comments and suggestions on our manuscript. We agree with most comments and modified/updated the manuscript accordingly. Below is a point-by-point reply, our answers appear in italics.*

This is an interesting study that compares 4 different methods for quantifying calcification rates under high and low pH conditions. The authors conclude that that alkalinity anomaly, Ca anomaly, and [45]Ca methods are all in close agreement, but the [13]C method is not. This is a helpful study for researchers that are trying to calculate calcification rates of individual corals. The methods are rigorous. However, I personally have only done the TA anomaly technique so hopefully the other reviewers have hands-on experience with the other 3 methods. My comments below are minor. I believe this will make a nice contribution to the coral biogeochemistry literature.

Abstract

Line 27: add a comma after calcification
> *Done*

Line 41: This is a bit of a meta comment, but what if the [13]C method is accurate and the other 3 are highly correlated, but wrong. How do we know which of these methods are "true" net calcification?
> *Interesting comment. The reason why we reject the [13]C method (as applied in our study) is not only because [13]C based rates are not correlated to the other methods but also because calcification rates based on this technique are much higher and much more variable than rates based on the other methods. As mentioned in the text, it is very unlikely that dissolution was a significant process during our incubations as nubbins were fully covered with tissue, therefore there is no distinction between net and gross calcification. Now, calcification (net or gross) consumes 1 mole of carbon and 1 mole of calcium to produce 1 mole of calcium carbonate. The fact that $\Delta[Ca]$ and $\Delta[A_T]$ and highly corelated following a 1:2 ratio fully confirms this. We should therefore have a 1:1 ratio between C and Ca fluxes, the fact that higher rates were obtained with the [13]C technique is problematic. Finally, several studies have shown that most of the calcium used by the calcification process comes from seawater, a significant proportion of the carbon used comes from the metabolism of the organism, suggesting that rates based on C incorporation ([14]C or [13]C) must significantly underestimate true net calcification.*

Introduction

Line 77: You can account for changes in nutrients (by measuring nitrate, phosphate, and ammonium and incorporating into the delta TA) as well as evaporation (normalize to salinity) in the alkalinity anomaly technique.
> *The reviewer is correct. We have added this small paragraph to deal with this comment: "This method assumes, however, that calcification is the only biological process influencing $A_T$ (Smith and Key, 1975). Nitrogen assimilation through photosynthetic activities, nitrification as well as aerobic and anaerobic remineralization of organic matter are known to impact $A_T$ through the consumption or release of nutrients (ammonium, nitrate and phosphate) and protons (Wolf-*

*Gladrow et al. 2007). While for some group of species (e.g. bivalves, sea urchins), corrections appear necessary to take into account the effect of nutrient release on $A_T$, changes in nutrient concentrations during incubations of isolated corals are too low (i.e. several orders of magnitude lower than changes in $A_T$) to introduce a significant bias in the calculations (Gazeau et al. 2015)."*

*Furthermore, ammonium concentrations have been measured at the start and end of selected incubations (only at ambient pH) that confirmed this assumption (D [NH$_4$] were at least 2 orders of magnitude lower than D$A_T$).*

*We do not discuss here the need to correct for evaporation as this is discussed in details later in the text.*

Line 96: Replace comma with semi-colon and add comma after "therefore".
   *Done*

Line 113 – 114: Incorporate this sentence into the last paragraph
   *Done*

Methods

Line 147: replace "a" with "and"
   *Done*

Line 180: remove "a"
   *Done*

Line 265 states that initial levels are not necessary to compute calcification and only final values with and without corals are used, but line 269 says that T1 are concentrations are the start of the incubations. This is a bit confusing. Please clarify.
   *Equations 3 and 4 present the calculation procedure showing that initial levels are not necessary to compute calcification rates as stated in the text above the equations. We believe it is important to detail these equations and do not believe this is confusing as presented. However, to make sure there is no misunderstanding we added: "where $A_{T1}$ and $Ca_1$ are $A_T$ and $Ca^{2+}$ concentrations at the start of the incubations (in $\mu$mol kg$^{-1}$; not used in the computations), …"*

Line 275 – 276: Please explain the parameters in the equations.
   *Done.*

Line 280: There is an empty box on the equation.
   *Corrected.*

I think it is worth discussing why different incubation times were used. Why not do them all at the same time to reduce error with changing carbonate chemistry in the background (i.e. the longest time needed to get a result from all 4 methods)?
   *We did not have this information before starting this study. Incubation times have been chosen based on practical aspects (access to the lab etc…). The fact that they differ between different incubations is not in conflict with our objective which was*

*to compare changes in various parameters during the same incubation, not to compare different incubations between each other. A sentence has been added in the Material and Method section: "Incubation times were not fixed based on scientific considerations and differed between the different incubations due to practical constrains (i.e. access to the lab etc…)."*

Please add incubation temperatures to table 1 or 2

*As temperature was maintained constant and at the same level for all incubations, the temperature level is now mentioned in the legend of both tables.*

Results section throughout: Instead of saying X and Y are presented in Figures 1 and 2, make a statement about the result and cite the figure and table after. (For example, see like 368).

*Modified accordingly.*

*We thank the reviewer for her/his comments and suggestions on our manuscript. We agree with most comments and modified/updated the manuscript accordingly. Below is a point-by-point reply, our answers appear in italics.*

This is a nice study comparing 4 different methods to measure short-term calcification rates in corals. The comparison of three less commonly used methods (calcium anomaly, [45]Ca, [13]C) with the commonly used alkalinity anomaly technique adds to the existing literature of method comparisons for estimating coral calcification. Furthermore, the current study has the benefit that the different methods were measured during the same incubation, minimizing the risk of other factors confounding the results. The authors show that two of the three methods are highly correlated and not significantly different from the alkalinity anomaly technique, and further provide useful recommendations on minimum and maximum incubation times for various volume to biomass ratios and techniques. Overall, this will be a useful addition to the existing literature on coral calcification methods. As a note of caution, I do not have experience with the calcium anomaly, [45]Ca and [13]C methods, therefore I cannot judge the experimental protocol used for these methods.

I only have one concern regarding the data: since there was no pH control during the incubations and some incubation times were rather long, especially when conducted in the dark, significant changes in carbonate chemistry did occur over the course of these incubations. For example, pH decreased from 8.05 to 7.62 under ambient conditions in the dark due to respiration and calcification. While this is clearly stated in the Results, the Discussion on acceptable changes in carbonate chemistry largely focuses on changes in delta $C_T$ rather than pH but I don't think such a change is acceptable in studies that actually aim to detect the impacts of low pH on coral calcification. Similarly, Riebesell et al. (2010) also recommend that changes in $A_T$ during incubations should be within 10% of starting $A_T$, yet changes in this study were typically larger than this, except under low pH. Furthermore, there is no discussion whatsoever regarding changes in dissolved oxygen and this was also not measured, despite hypoxia/hyperoxia potentially stressing the corals. Again, while this may be less relevant for a method comparison, it is certainly relevant when making recommendations for general incubation times. I would therefore encourage the authors to discuss these aspects in more detail in the Discussion.

> *Many thanks for these very constructive comments. As stated in the manuscript (but clarified in the revised version), our study was designed to compare different techniques to estimate calcification rates and not to define the best experimental approach to study the effects of ocean acidification on coral species using these different approaches. As such, the chosen experimental protocol (e.g. incubation times) was not optimal and led, in some cases, to significant changes in the carbonate chemistry during incubations. We fully agree with the reviewer that the method we used to estimate maximal incubation times (i.e. only implying a change in $C_T < 10\%$) is not acceptable. Indeed, as stated by the reviewer, one should not only focus on $C_T$ but on pH and $A_T$ as well in order to make sure that carbonate chemistry is maintained under an acceptable range (as compared to starting conditions). While we could find in the literature some estimates of "acceptable" changes in $C_T$ and $A_T$ (respectively Langdon et al., 2010 and Riebesell et al., 2010), it is more difficult to estimate what changes in pH are acceptable. As such, we have arbitrarily decided to consider a maximal change in pH set to 0.06 which is the minimal change in global surface ocean pH projected for 2100. Therefore, the new estimated tmax corresponds to the lowest value between tmax_pH ($\Delta pH_T < 0.06$), tmax_$C_T$ ($\Delta C_T < 10\%$) and*

*tmax  $A_T$ ($\Delta A_T$ < 10%). Except in the light under ambient pH conditions, tmax is always set to the maximal incubation time allowed to keep pH levels under an acceptable range ($\Delta pH_T$ < 0.06).*

*Regarding oxygen levels, as pointed out by the reviewer, oxygen levels were not measured. However, our incubations were conducted in continuously mixed open systems, allowing equilibration with the atmosphere. Exchange at the air-sea interface is considerably faster for $O_2$ than for $CO_2$. Furthermore, we have unpublished data from an other experiment that confirm that under the same experimental setup, where we also tracked the dissolved oxygen concentration over time, we did not observe any significant deviation from saturation.*

*The new paragraph now reads:*

*"Our study was designed to compare different techniques to estimate calcification rates and not to define the best experimental approach to study the effects of ocean acidification on coral species using these different approaches. As such, the chosen experimental protocol (e.g. incubation times) was not optimal and led, in some cases, to significant changes in the carbonate chemistry during incubations. However, our results provide some insights that we further discuss in the following section. Measuring and comparing calcification rates of organisms under varying pH conditions requires the careful choice of a volume and a time interval such that the precision of the calcification rate measurement is large enough to observe significant signals and that the change in carbonate chemistry parameters between the beginning and end of the incubation is small compared to the range of these parameters in the different treatments (Langdon et al. 2010). Table 5 illustrates the incubation time necessary to obtain measurable changes for each method ($t_{min}$) considering the ratio between incubation volume and coral size chosen for our study. As the $^{13}C$ incorporation method did not provide reliable rates, this technique was not considered in this analysis. The threshold for significant signals was set at 10-fold the analytical precision of the instruments (Langdon et al. 2010) for $A_T$ and $Ca^{2+}$ measurements (1.2 and 2.9 µmol $kg^{-1}$, respectively) and above the detection limit of 15 cpm for $^{45}Ca$ activity estimated. Maximum incubation times are more difficult to estimate. Langdon et al. (2010) and Riebesell et al. (2010) recommend considering incubation times short enough to maintain $A_T$ and $C_T$ within an acceptable range ($\Delta A_T$ and $\Delta C_T$ < 10%). As it is more difficult to estimate what changes in pH are acceptable, we have arbitrarily considered a maximal change in pH of 0.06, corresponding to the lowest change in global surface ocean pH projected for 2100 (IPCC, 2014). Maximal incubation times, as presented in Table 5 ($t_{max}$), correspond then to incubation times that should not be exceeded in order to maintain acceptable conditions of the carbonate chemistry ($\Delta pH_T$ < 0.06 and $\Delta A_T$ < 10% and $\Delta C_T$ < 10%).*

*Under light and ambient pH conditions, even if the ratio between incubation volume and nubbin size is much higher than for previous similar studies (e.g. Cohen et al. 2017), all methods would allow a precise estimation of calcification rates over very short incubation times (~15 min to 1 h, depending on the method) while leading to moderate changes in carbonate chemistry. In the dark, and under ambient pH conditions, in the absence of pH increase due to photosynthesis, the decrease of pH due to respiration, narrows the possible incubation period to 1.3 h. While this is still larger than the incubation time allowing to obtain a significant signal with alkalinity*

*anomaly technique (~20 min), the other two methods necessitate longer incubation times to obtain precise estimates (> 1.5 h). At lower pH, both under light and dark conditions, and using open systems without a continuous pH regulation as in our study, it is obvious that all techniques are not well adapted to this experimental protocol. Indeed, as a consequence of lower calcification rates at lower pH and significant $CO_2$ degassing, incubation times necessary to obtain significant signals using these techniques are too large to maintain the carbonate parameters within an acceptable range. This is not insurmountable as a continuous regulation of pH using for instance pure $CO_2$ bubbling or incubations performed in a closed container (i.e. without contact to the atmosphere) would alleviate these problems.*

**Specific Comments**

Abstract

L32: please state the respective pH values instead of ambient and low
  *Added*

Introduction

L61: please also cite here other studies that recently compared various calcification methods, such as (Gazeau et al. 2015), (Schoepf et al. 2016) and (Cohen et al. 2017)
  *References added.*

L84: "solid agreement" – this is rather colloquial and should be rephrased, e.g. "good agreement"
  *Modified*

L114: you could add here that this was done under different pH and light conditions
  *Added*

Methods

L124-138: Please provide more information on how water motion/flow was provided in the aquaria, how big the tanks were, rate of seawater renewal etc
  *Now provided.*

L127: please provide more information on how many branches from how many different parent colonies were collected for each experiment.
  It now reads: "*In June 2017, 40 terminal portions branches of S. pistillata, free of boring organisms, were cut from four different parent colonies (10 branches per parent colony) and suspended by nylon lines to allow tissues to fully cover the exposed skeleton for at least five weeks (Tambutté et al., 1995; Houlbrèque et al., 2015).*"

L130: what was the concentration of Artemia fed during experiment 1? This info is only provided for experiment 2

*Added.*

L137: please change to "biometrics parameters of the biological material"
*Modified.*

L146: looking at Fig. 1, I wonder whether the rod to which the nylon line was attached shaded the coral from light coming from above?
*The thickness of the holder was only 4 mm. The position of the lights and water movement inside the incubation chamber allowed nubbins to slowly move inside the chamber and ensured no significant shading.*

L147: should be "and low pH"
*Corrected.*

L273: a description of how coral skeletal dry weight was measured is missing from the Methods. Please add.
*This was mentioned in the text, we added the apparatus used to weigh the samples: "Tissues were then dissolved completely in 1 mol $L^{-1}$ NaOH at 90 °C for 20 min. The skeleton was rinsed twice in 1 mL NaOH and twice in 5 mL in MilliQ water. It was then dried for 72 h at 60 °C, precisely weighed at ± 0.01 g using a Sartorius BP 310S (referred thereafter to as skeleton dry weight), and dissolved in 12 N HCl."*

L309: It's good to see that model II regressions were used for the analyses.
*Thanks, this is indeed appropriate when both variables are associated to experimental errors.*

Results

L313: Table 2: why was the seawater activity much higher in experiment 2 than 1?
*Added in the text (line 170): As we anticipated lower calcification rates during the set of experiments conducted at low pH, initial nominal activity was set to ~30 Bq $mL^{-1}$.*

L316: please state whether this is SD or SE
*Since we present SD values for all environmental conditions (as opposed to SE when we refer to estimated rates), a sentence has been added at the start of the Results section: "All values in Table 2 as well as in the text below correspond to the average between replicates (or incubations) ± standard deviation (SD)."*

L328: was this change in pH during incubation similar for the different methods?
*Indeed, as mentioned in the text, changes in pH were similar for the different incubations. Final pH levels were:*
- *In the light*
  - $^{45}Ca$: $pH_T$ (8.05 ± 0.03; n = 6)
  - $^{13}C$: $pH_T$ (8.06 ± 0.04; n = 6)
- *In the dark*
  - $^{45}Ca$: $pH_T$ (7.61± 0.1; n = 6)
  - $^{13}C$: $pH_T$ (7.63 ± 0.04; n = 6)

L336: should be "were similar"

*Corrected*

L361: there are also some other data with asterisks in Table 3 – I assume they are also outliers but this is not explicitly discussed. Please clarify.
> *Clarified: "These estimates (n = 4) have been considered as outliers, marked with an asterisk in Table 3 and not included in the following analyses."*

Discussion

L443: please replace "that" with "why"
> *Replaced.*

L461: should be "was" x2
> *Modified.*

L492: should be "importantly"
> *Modified to "significant".*

L514: would be necessary for what? Please add.
> *Modified to: "Conducting similar comparison studies with other coral species as well as other major calcifying groups widely studied in the context of ocean acidification (e.g. coralline algae, molluscs etc…) would be necessary **for a better understanding of ocean acidification impacts on ecosystem services provided by calcifying organisms.**"*

Figures and Tables

Table 3 is very long. I think this information could be better represented in a figure showing both the average of all six replicates per treatment/method and the individual data points spread around the average.
> *We respectfully disagree and prefer keeping the table as it is, as we believe it is important to provide the actual numbers to the reader. Individual data points are further shown in Figures 2-4.*

Also, the legend does not currently explain what the asterisk next to some data means. Please add.
> *Added.*

Table 4: please add the p-value for the regressions to the table.
> *Added.*

[revised manuscript text omitted]